



# Amine and guanidine emissions from a boreal forest floor

Marja Hemmilä[1], Ulla Makkonen[1], Aki Virkkula[1,2], Georgia Panagiotopoulou[1], Juho Aalto[2], Markku Kulmala[2], Tuukka Petäjä[2], Hannele Hakola[1], Heidi Hellén[1]

[1]Finnish Meteorological Institute, P.O. Box 503, 00101 Helsinki, Finland

[2]Insitute for Atmospheric and Earth System Research / Physics, Faculty of Science, University of Helsinki, Finland

*Correspondence to*: Marja Hemmilä (marja.hemmila@helsinki.fi)

**Abstract.** We measured amine and guanidine emission rates from a boreal forest floor in Finland with 1-h time resolution, using an online ion chromatograph (instrument for Measuring AeRosols and Gases in Ambient air – MARGA) coupled with an electrospray ionization-quadrupole mass spectrometer (MS). MARGA-MS was connected to a closed dynamic flow-

through poly(methyl methacrylate) chamber. Chamber recovery for the emission measurements was tested semi-quantitatively for monomethyl-, dimethyl- and trimethylamine (MMA, DMA and TMA), and the results were 19%, 29% and 24%, respectively. MMA, DMA and TMA showed maximum emission rates in July, but the highest emission rates for guanidine were in April, when snow was melting. The MMA, DMA and TMA emission rates also clearly varied diurnally, especially in July with maxima at afternoon. Diethylamine (DEA) also showed higher emission rates, with clear diurnal cycles in July.

Other amine emission rates were mostly below the detection limits.

The temperature dependencies of the emissions were studied, and we noted a correlation between the emission rates and chamber temperature ($T_{chamber}$). Especially in July emission rates of DMA followed $T_{chamber}$ measured two hours earlier and guanidine showed a similar pattern. On the other hand, the TMA emission rates correlated with $T_{chamber}$ measured at the same time. This could be due to lower vaporizing temperature of TMA. Emission rates of DMA and TMA showed some air

temperature ($T_{air}$) dependency, but for MMA dependency was not as clear.

## 1 Introduction

Atmospheric aerosols are important to the climate, because they absorb and scatter solar and thermal radiation, and act as cloud condensation nuclei (IPCC 2013). Amines have been suggested to be key compounds in the secondary aerosol formation by both models (Kurten et al. 2008; Paasonen et al. 2012) and laboratory tests (Angelino et al., 2001; Petäjä et al., 2011; Yu et

al., 2012; Almeida et al., 2013; Glasoe et al., 2015). Amines are gaseous bases, whose general formula is $NR_3$, where $R$ denotes hydrogen or alkyl or aryl group. Since gas-phase amines cluster efficiently with atmospheric acid clusters (such as sulphuric acid, Kurtén et al. 2011) and therefore participate in neutralization in the atmosphere, it is difficult to detect their true atmospheric concentrations. Gas-phase amine concentrations from boreal forest air have been measured in a few studies (Sellegri et al., 2005; Kieloaho et al., 2013; Kulmala et al., 2013, Sipilä et al., 2015; Hemmilä et al., 2018). In these studies,

the observed alkylamine concentrations ranged from below the detection limit to ~150 $ppt_v$, depending on the sampling time



and the analysis method used. The main known anthropogenic sources of amines globally are animal husbandry, industry and composting processes, while natural sources are assumed to be oceans, biomass burning, vegetation and soil (Ge et al., 2011; Sintermann et al. 2014). Degradation of organic nitrogen compounds, especially carboxylation of amino acids, may be a source of low-weight alkylamines in soils (Yan et al., 1996). Under aerobic conditions, e.g. proteins, carnitine and choline in soil,

could be degraded to trimethylamine (TMA) and further dimethyl- (DMA) and methylamines (MMA) (Rappert and Müller 2005). Amines from soil can probably enter the atmosphere via volatilization (Ge et al., 2011).

However, direct flux measurements of alkylamines are difficult to perform and are very rarely done (Sintermann and Neftel, 2015), due to the high reactivity of amines and the lack of suitable measurement techniques. Amines also are 'sticky', so they are easily lost in the inlets of instruments. In our previous study (Hemmilä et al., 2018) we developed a method for

measuring atmospheric amines, using an in situ ion chromatograph (IC) connected to a mass spectrometer (MS), and we measured the ambient concentrations of different amines at the boreal forest site. We found that melting snow and soil may be potential sources of amines, especially MMA and TMA. Kieloaho et al. (2017) estimated the magnitudes of soil-atmosphere fluxes of DMA and diethylamine (DEA), using a gradient-diffusion approximation based on measured concentrations in soil solution and in the canopy air space. They found that boreal forest soil is a possible source of DMA and a sink for DEA. The

idea of melting snow and soil as amine sources and also the study by Kieloaho et al. (2017) inspired us to measure boreal forest floor emissions.

Other strong bases can also be relevant to aerosol formation. For example, in a recent model study, the role of a strong organobase, guanidine, was examined in a sulphuric acid-driven new-particle formation (Myllys et al., 2018). The authors concluded that much less guanidine is needed for efficient particle formation than for DMA, which explains why we included

guanidine in the study. Guanidine is a catabolite of arginine and has been found in urine (Marescau et al. 1992, Van Pilsum et al., 1956). Arginine concentrations have been detected in a boreal forest in Alaska, USA (Werdin-Pfisterer et al., 2009).

The aim here was to examine amine and guanidine emissions from melting snow and from the boreal forest floor, using a closed dynamic flow-through chamber during different seasons. This method has been commonly used for studying biogenic volatile organic compounds (BVOCs), especially mono- and sesquiterpene and isoprene emissions from boreal forest

soil (Hellén et al., 2006; Aaltonen et al., 2011; Mäki et al., 2017).

## 2 Experimental

### 2.1 Measurement site

Forest floor emission measurements were performed at SMEAR II (Station for Measuring Forest Ecosystem–Atmosphere Relations II) station in Hyytiälä, southern Finland (61o510′N, 24o170′E, 180 m a.s.l., Hari and Kulmala, 2005) from March

to September 2018 about 1 week per month. The forest stand at the SMEAR II station is ~60 years old, ~19m height and dominated by Scots pine (*Pinus sylvestris* L.) (75 % of stand basal area). Also, some Norway spruce (*Picea abies* (L) H. Karst), aspen (*Populus tremula* L.) and birch (Betula L.) also grow in the forest. The most common vascular plant species at ground



level are lingonberry (*Vaccinium vitis-idaea* L.), bilberry (*Vaccintum myrtillus* L.), wavy hair-grass (*Deschampsia flexuosa* (L.) Trin.) and heather (*Calluna vulgaris* (L.) Hull.), and the most common mosses are Schreber´s big red stem moss

(*Pleurozium schreberi* (Brid.) Mitt.) and dicranum moss (*Dicranum Hedw*.) (Ilvesniemi et al, 2009). The soil above the homogenous bedrock is Haplic podzol on glacial till.

## 2.2 Measurement method

The emissions were measured, using a flow-through technique with a stainless-steel collar, a poly(methyl methacrylate) (PMMA) chamber (60 cm * 60 cm * 80 cm) and   12-m-long heated fluorinated ethylene propylene (FEP) tubing.

Polytetrafluoroethene and stainless steel were also tested as chamber materials, but PMMA was the most useful.  The collar was installed in autumn 2011 close to our container, where the analytical instrumentation was located. Soil surface coverage of the chamber area was determined by visual inspection and photographs (see Fig. S1). Based on this analysis the chamber area included litter, lingonberries, bilberries and a few chickweed-wintergreens (*Trientalis europaea* L.). Samples from the chamber air were directed to MARGA (Monitor for AeRosols and Gases in Ambient air (Metrohm-Applikon, Schiedam, The

Netherlands) (ten Brink et al., 2007), which is an online-IC. Makkonen et al.  measured inorganic gases and aerosols with the MARGA instrument, both in urban (2012) and rural (2014) environments. In addition, the MARGA system was connected to an electrospray ionization-quadrupole MS (Shimadzu LCMS-2020, Shimadzu Corporation, Kyoto, Japan) to improve the sensitivity of the amine measurements (Hemmilä et al., 2018).  Since July, we also used $PM_{10}$-cyclone (URG 1032, Teflon-coated) with the inlet tubing.

We measured six different amines and guanidine; the amines included MMA, DMA,  TMA, ethylamine (EA), DEA and propylamine (PA), and the detection limits for the analysis system of MARGA-MS were 1.0, 13.4, 14.2, 1.8, 1.2 and 1.6 ng $m^{-2}$ $h^{-1}$, respectively for the amines, and 3.4 ng $m^{-2}$ $h^{-1}$ for guanidine. Calibration for the system was done one to two times per month; see details from Hemmilä et al. (2018). The emission rate (E) was calculated from the MARGA-MS results with equation 1 as follows:


$$E = \frac{c*F_{in}}{A} \tag{1}$$

where c is the measured concentration (ng $m^{-3}$), Fin is the flow to the chamber ($m^3$ $h^{-1}$) and A is the enclosed forest floor area ($m^2$).

For more detailed description of the amine analysis method, see Hemmilä et al. (2018). Guanidine was included in the previous method, since it is an efficient compound for new particle formation (Myllys et al., 2018). The sampling time was 1 h, sample flow rate was 16.7 l $min^{-1}$ and the sampled air was replaced with amine-free air (flow rate 25 l $min^{-1}$).  We used an oxalic acid filter to remove bases from the air that went into the chamber.



The functionality of the chamber was tested with a permeation oven. In this process, air containing 501, 630 and 3005

ng m$^{-3}$ of MMA, DMA and TMA, respectively, was flushed into the chamber at a flow rate of 25 l min$^{-1}$. The concentrations

inside the chamber were measured with MARGA-MS and the 12-m long heated FEP inlet tubing. We found that the recoveries

of the MMA, DMA and TMA were only 19%, 29% and 24%, respectively, so wall losses were significant in the chamber. The

FEP and stainless-steel chambers were also tested, but with the FEP-chamber recovery was poorer and the stainless-steel

chamber had a too high background level.  Since the recovery test results were only semi-quantitative and made only for MMA,

DMA and TMA, we did to not take them into account when showing the emission measurement results. Therefore, the true

emissions were expected to be higher than as found in this study.

## 2.3 Meteorological conditions

The meteorological quantities for the chamber system included measurement of relative humidity (RH) sensors (Honeywell

HIH-400; Honeywell International Inc., Charlotte, NC, USA and Rotronic Hygroclip XD; Rotronic AG, Bassersdorf,

Switzerland), temperature (T) sensors (TI LM35; Texas Instruments, Dallas, TX, USA and Rotronic Hygroclip XD; Rotronic

AG ), photosynthetically active radiation (PAR) sensors (Apogee SQ-520; Apogee Instruments Inc., Logan, UT, USA and a

Li-cor LI-190R quantum sensor; LI-COR Biosciences Inc., Lincoln, NE, USA). The ambient meteorological quantities for the

site were obtained from the SmartSmear AVAA portal (Junninen et al., 2009). SmartSmear is the data portal used for

visualization and downloading of continuous atmospheric, flux, soil, tree, physiological and water quality measurements at the

SMEAR research stations of the University of Helsinki.

## 3 Results and discussion

### 3.1 Emissions of different amines and guanidine

We measured the amine and guanidine emissions from the boreal forest floor in April, May, July and September in 2018. In

Fig. 1 the average emissions in every month are shown. MMA, DMA, TMA, and DEA showed maximum emissions in July,

but guanidine already showed the highest emissions in spring. In our previous ambient air study (Hemmilä et al., 2018) we

observed that DMA and TMA also showed their maximum ambient air concentrations in July. In that study, MMA showed

maximum ambient air concentrations in spring, but here the highest emission rates from the forest floor were measured in

summer. The MMA still showed higher emission rates in April than did DMA or TMA, which could be explained by its lower

boiling point, so it vaporizes more readily. Sarwar et al. (2005) measured ammonia emissions in a pine forest in Texas and

obtained an average result in the summer months of 0.09 kg km$^{-2}$ per month or approximately 12 500 ng m$^{-2}$ h$^{-1}$, which is about

35 times higher than our maximum average DMA emissions (320 ng m$^{-2}$ h$^{-1}$). In taking into account our measurements, 29%

recovery for DMA, the emission rates measured by Sawar et al. were about 10 times higher than ours. Figures 2 and 3 show

the diurnal variations in photosyntetically active radiation (PAR), chamber temperature (T$_{chamber}$), soil surface temperature

(T$_{soil}$) and soil surface humidity (SH) during measurement periods in April, May, July and September. The diurnal cycles for





the environmental conditions and amine and guanidine fluxes were expressed as hourly means on a monthly basis. Due to the
MARGA-MS instrument features, some data were shown in zigzag patterns; for more information, see Hemmilä et al. (2018).

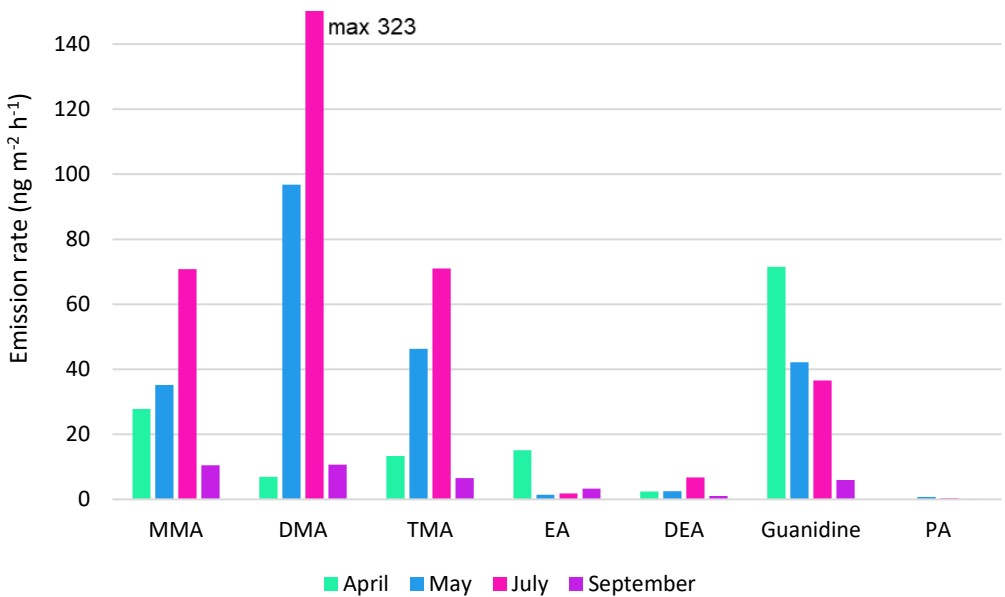

**Figure 1: Average monomethyl- (MMA), dimethyl- (DMA), trimethyl- (TMA), ethyl- (EA), diethyl- (DEA) and propylamine (PA)**
**and guanidine emission rates (ng m⁻² h⁻¹) in different months.**

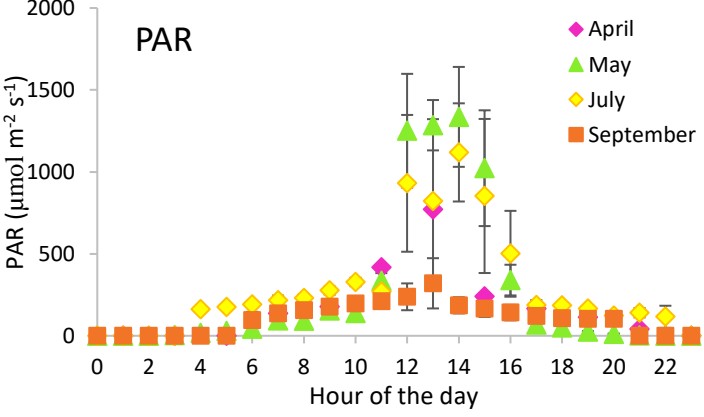

**Figure 2: Diurnal variation in photosynthetically active radiation (PAR) during measurement periods in April, May, July and**
**September. The error bars show the standard deviation between the measured values.**



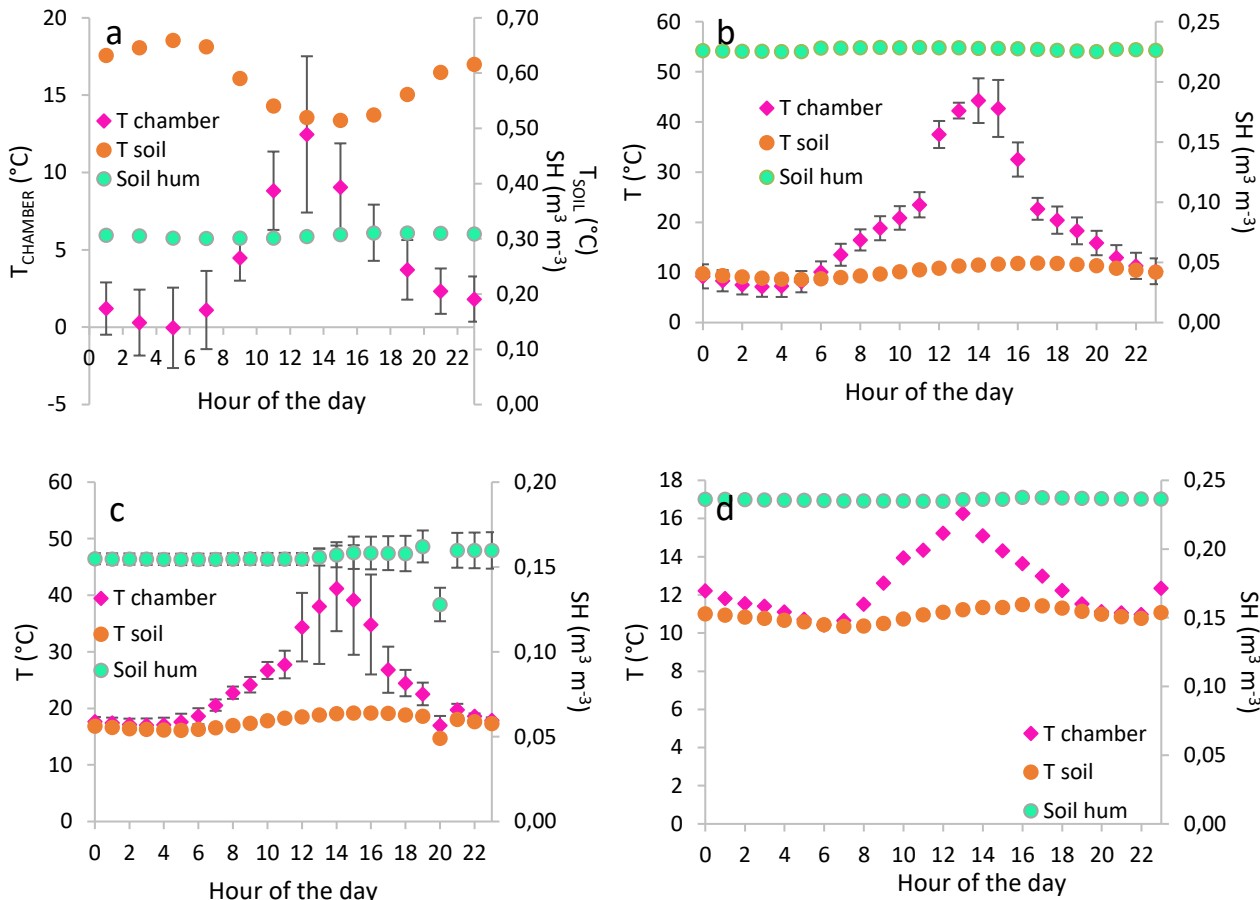

**Figure 3: Diurnal variations in chamber (T$_{chamber}$) and soil surface temperature (T$_{soil}$) (°C) and soil surface humidity (soil hum) (m$^3$ m$^{-3}$) during measurement periods in April (a), May (b), July (c) and September (d). The error bars show the standard deviation between the measured values. In September, adding the error bar changed the scale so that the variations in diurnal cycle were blurred. Note the difference in emission rate axis length between the figures.**

Figure 4 shows the mean diurnal variation in emission rates of MMA in April, May, July and September 2018. The emission rates of MMA in April were quite low (~30 ng m$^{-2}$ h$^{-1}$) with no clear diurnal variation, but in May and even more so in July, they increased and showed clear diurnal variation with maxima in the late afternoon and minima in the early morning. In September, the emission rates of MMA were mostly under the detection limits, but again the highest emission rates were in the late afternoon.





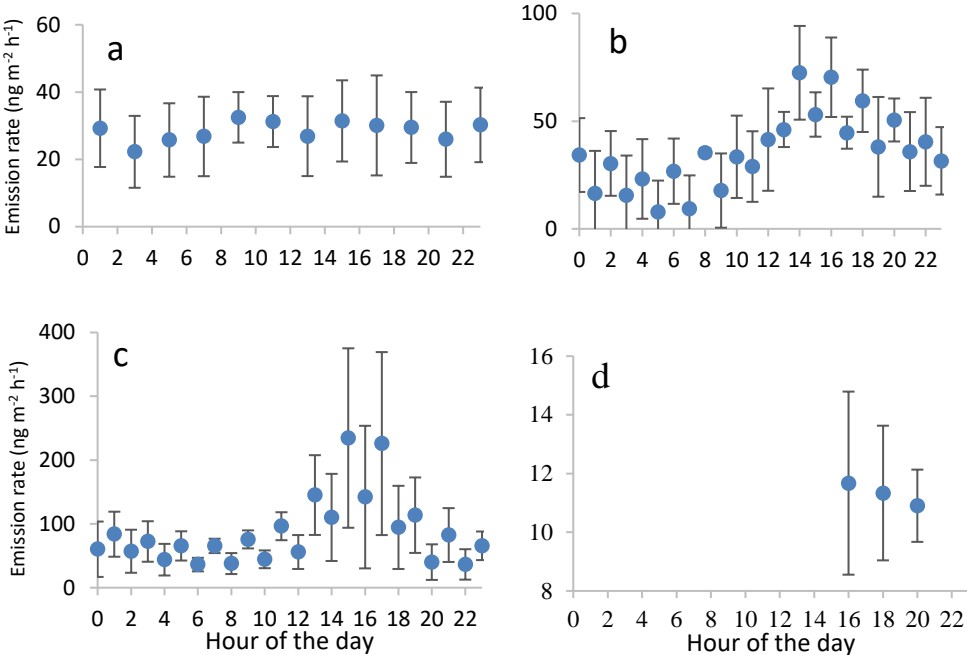

**Figure 4: Diurnal variations in monomethylamine emission rates in April (a), May (b), July (c) and September (d). The error bars show the standard deviation between the measured values. Note the difference in emission rate axis length between the figures.**


The DMA showed clear diurnal cycles (Fig. 5) in every measurement period, with maxima in the afternoon. In spring, the emission rates were quite modest, but became more intense as summer proceeded. The highest emissions for DMA were in July, when the average afternoon maximum was over 1000 ng $h^{-1}$ $m^{-2}$. In September, the emission rates decreased to even lower values than in April, but they still showed maxima in the afternoon. Kieloaho et al. (2013) also measured at the same

site high DMA+EA concentrations in the ambient air during July, but even higher ambient concentrations were measured in autumn. Even though both measurements were conducted at the same site, there were more possible sources, meteorological conditions and removal mechanisms contributing to ambient air concentrations and this may explain the difference in behaviour. Based on the 2013 data and a model, Kieloaho et al. (2017) concluded that boreal forest soil is a source of DMA, and our measurements confirmed this.

The DEA emission rates were generally quite low and diurnal variability was only detected in July with maxima (~ 30 ng $h^{-1}$ $m^{-2}$) in the afternoon (Fig. 6). Our measurements agree with those of Kieloaho et al. (2013), who also detected the highest DEA concentrations in the ambient air in summer. However, Kieloaho et al. (2017) focused on DEA soil flux dependence on different physical and chemical state variables and noted, that the flux estimates were especially sensitive to soil pH changes. The typical pH of boreal forest soil was 5.3, and the compensation point of DEA was 5.7. Based on this,

Kieloaho et al. concluded that boreal forest soil could be a possible sink for DEA, whereas our afternoon maxima suggested soil emissions. The soil pH was not measured, so we do not know if this could have caused the discrepancy.





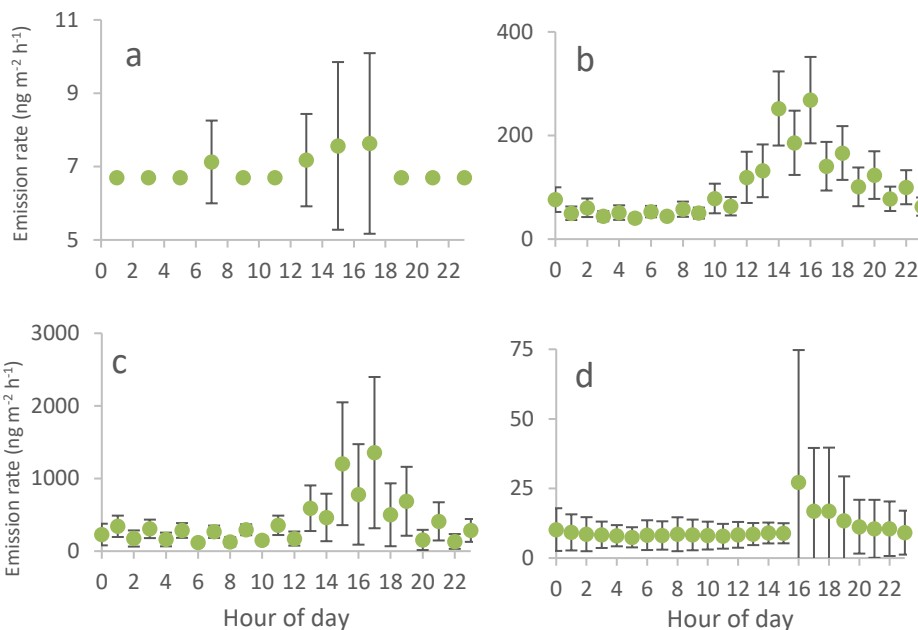

Figure 5: Diurnal variations in DMA emission rates in April (a), May (b), July (c) and September (d). The error bars show the standard deviation between the measured values. Note the difference in emission rate axis length between the figures.

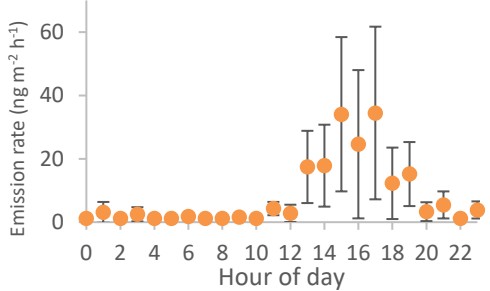

Figure 6: Diurnal variations in diethylamine emission rates in July. The error bars show the standard deviation between the measured values.

The TMA, similar to DMA, showed clear diurnal cycles in every measurement period (Fig. 7). The emission rates were similar to those of DMA in April and September, but in May (maximum ~80 ng h$^{-1}$ m$^{-2}$) and July (maximum ~300 ng h$^{-1}$ m$^{-2}$) they were significantly lower than the DMA emission rates. However, the emissions of TMA were higher than the MMA emissions in July. The TMA and MMA seemingly showed maxima earlier in the afternoon than DMA, which could be explained by their more ready vaporization.


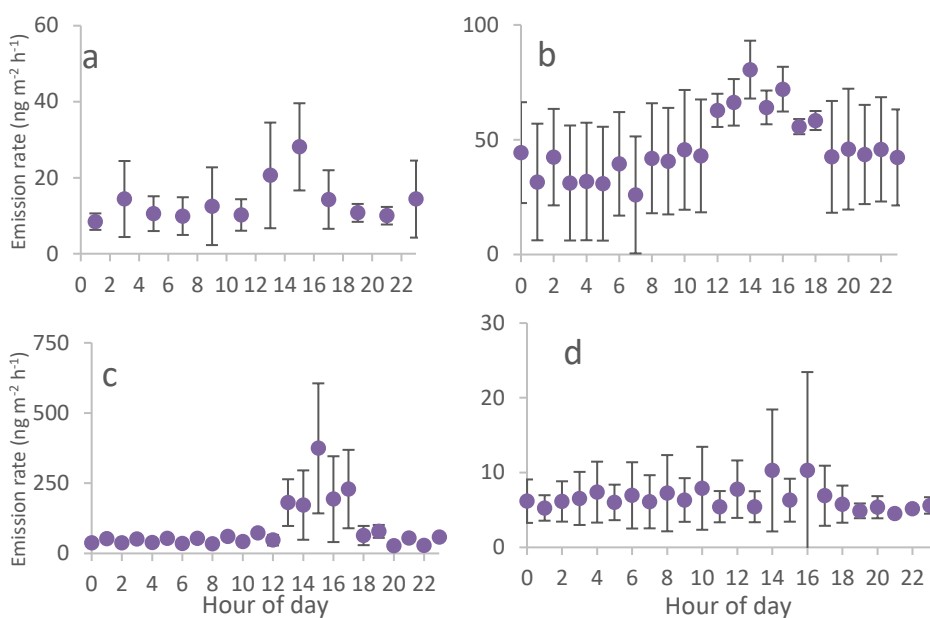

**Figure 7: Diurnal variations in trimethylamine emission rates in April (a), May (b), July (c) and September (d). The rror bars show the standard deviation between the measured values. Note the difference in emission rate axis length between the figures.**


Guanidine did not show as clear a diurnal variation as did MMA, DMA or TMA (Fig. 8). Unlike the other compounds, guanidine showed the highest emission rates in April, which could indicate it was trapped in snow.



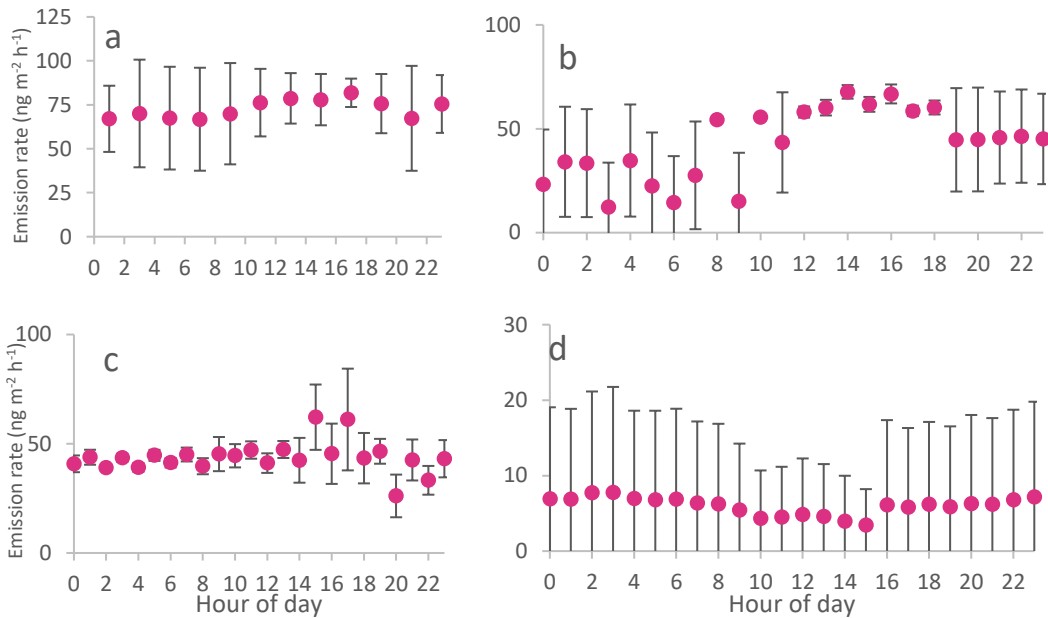

**Figure 8: Diurnal variations and error bars of guanidine emission rates in April (a), May (b), July (c) and September (d). The error bars show the standard deviation between the measured values. Note the difference in emission rate axis length between the figures.**

### 3.2 Temperature dependency of emissions

In May and July, the diurnal variation in amine emission rates, especially for DMA, followed chamber temperature (Fig. 9a and b). The $T_{chamber}$ dependence was best shown when the DMA and guanidine emissions were plotted against $T_{chamber}$ measured two hours earlier (Fig. 10, Table 1). Due to the large size of the chamber compared with that of the flow going in, it requires time before the concentration stabilizes to the new level of the emissions. However, for TMA we did not detect as clear a difference, possibly because TMA evaporates more readily than DMA. It may also require possible some time for the DMA emissions to react to the temperature changes of the chamber air. The large size of the chamber could have resulted an hour's delay in the emission rates. For example, the temperature of soil surface layer, which could be a source of these compounds, is expected to follow ambient air temperature changes slowly behind. This also indicates that soil could be the main source of DMA and guanidine, but for TMA the source could be surface vegetation, which respond more readily to the changes in the air temperature.

For DMA $T_{chamber}$ dependency was better illustrated with an exponential curve and for TMA and guanidine with a linear curve. Table 1 shows that the daily emission potentials at 30 ºC for DMA were clearly higher in July than in May, whereas the emission potentials of TMA and guanidine remained similar in both months. The $T_{chamber}$ and the PAR were well correlated in May and July. Unfortunately, the RH data of the chamber were not collected during July. We believe, that the





higher DMA emission potential in July could have been due to higher activity in the soil processes. The mean monoterpene emission potential (at 30 ℃) in July of the forest floor at the same site was 6.44±7.54 µg m$^2$ h$^{-1}$ and sesquiterpene 0.15±0.29 µg m$^2$ h$^{-1}$ during the growing season in 2015 (Wang et al., 2018). These emission potentials for monoterpenes were 10–230 times higher and for sesquiterpenes 0.2–5 times higher than our calculated emission potentials of amines during May and July 2018. The measured recovery from our chamber was ~25% for amines; therefore, the true emission was expected to be four

times higher, but still clearly lower than the monoterpene emissions. In comparison to the emission rates sesquiterpene, the fixed amine emission rates were probably similar.

Even though the diurnal variation in DMA followed the $T_{chamber}$ quite well, there were wide variations in the emission rates of different days. For example, in July the emission rates suddenly dropped in the 20$^{th}$ and more dramatically in the 21$^{st}$. Clearly, other important factors affected the emissions. We compared our emission rate data with additional data and noted,

that the PAR values were also lower on the 20$^{th}$ and 21$^{st}$ and it was slightly rainy in the evening of 20$^{th}$ and in the early afternoon of 21$^{st}$. The soil surface water content also began increase in the evening of 20$^{th}$. Amines are water-soluble and can therefore be flushed away during the rain or remain on the wet surfaces.

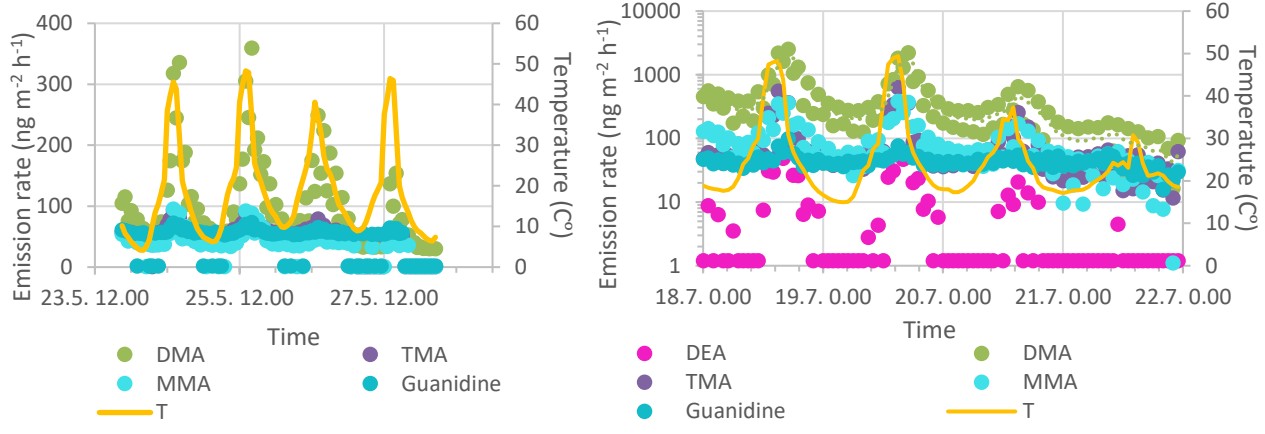


**Figure 9: Time series of amine and guanidine emission rates (ng m$^{-2}$ h$^{-1}$) and temperature of the chamber (°C) in May and July 2018.**





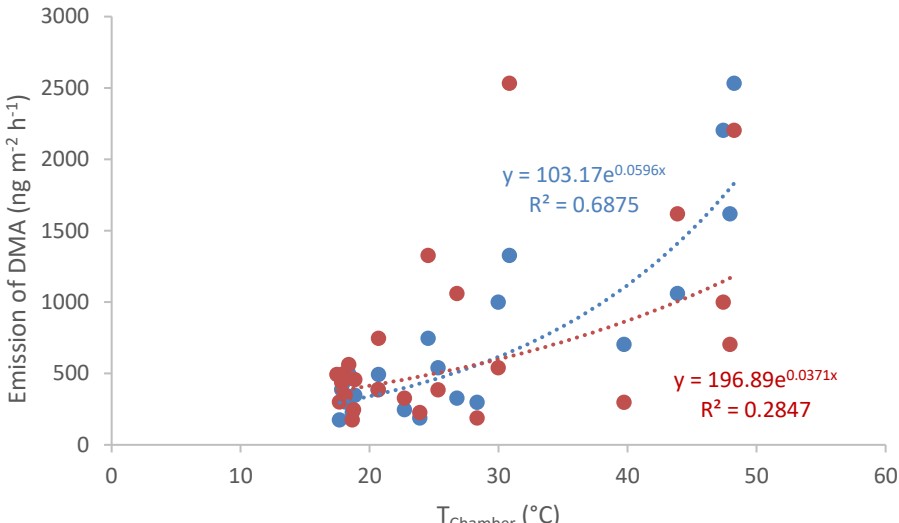

**Figure 10: Emissions of DMA 18.7.2018 compared with chamber temperature at the same time (red) and chamber temperature two**
**hours earlier (blue).**

**Table 1: Calculated emission potentials of DMA (exponential function), TMA and guanidine (linear function) at temperature of 30 ºC on different dates. The red values mark that the coefficient of determination ($R^2$)-value was low. Other curve parameters are in Supplement Table ST1.**

| | DMA | | | TMA | | Guanidine | | |
|---|---|---|---|---|---|---|---|---|
| Date | Emission potential (ng m$^{-2}$ h$^{-1}$) | $R^2$ | $R^2$ without temperature move | Emission potential (ng m$^{-2}$ h$^{-1}$) | $R^2$ | Emission potential (ng m$^{-2}$ h$^{-1}$) | $R^2$ | $R^2$ without temperature move |
| 24.5. | 161 | 0.86 | 0.64 | 68 | 0.48 | 61 | 0.28 | 0.18 |
| 25.5. | 152 | 0.86 | 0.64 | 65 | 0.72 | 59 | 0.35 | 0.35 |
| 26.5. | 149 | 0.70 | 0.50 | 63 | 0.58 | 59 | 0.19 | 0.11 |
| 27.5. | 69 | 0.73 | 0.38 | 36 | 0.47 | 42 | 0.37 | 0.41 |
| 18.7. | 617 | 0.69 | 0.28 | 147 | 0.58 | 51 | 0.61 | 0.15 |
| 19.7. | 506 | 0.78 | 0.41 | 168 | 0.66 | 49 | 0.68 | 0.56 |
| 20.7. | 277 | 0.16 | 0.15 | 102 | 0.31 | 45 | 0.057 | 0.097 |
| 21.7. | 43 | 0.05 | 0.0055 | 33 | 0.0047 | 28 | 0.17 | 0.0068 |


The emission rates of amines were also compared with $T_{chamber}$ and chamber relative humidity $RH_{chamber}$ (Fig. 11a) measured at the same time. With higher $RH_{chamber}$ and lower $T_{chamber}$, the emissions of MMA and DMA were lower, while the TMA emissions did not express the same pattern. In Fig. 11a, the highest measured emissions are missing





because our $RH_{chamber}$ sensor was unfortunately out of order in July. In Fig. 11b, the emission rates of MMA, DMA

and TMA were compared with $T_{air}$ and air relative humidity ($RH_{air}$) measured at the same time. For DMA and
TMA, the emission rates showed some $T_{air}$ dependency, but for MMA the dependency was not as clear ($R^2 = 0.$
46, $R^2 = 0.47$ and $R^2 = 0.26$ for DMA, TMA and MMA, respectively). From the exponential function $y = E_0 exp(kt)$,
it can be seen that $k_{DMA}$ is about two times that of $k_{TMA}$ and about 15 times that of $k_{MMA}$, suggesting that the DMA
emissions are two times more sensitive to $T_{air}$ changes than the TMA emissions and about 15 times more sensitive

than the MMA emissions. The emission rates did not seemingly show any dependency of $RH_{air}$. In our previous
study (Hemmilä et al., 2018) we noted that gas-phase DMA concentrations showed $T_{air}$ dependency ($R^2 = 0.55$),
confirming these results.

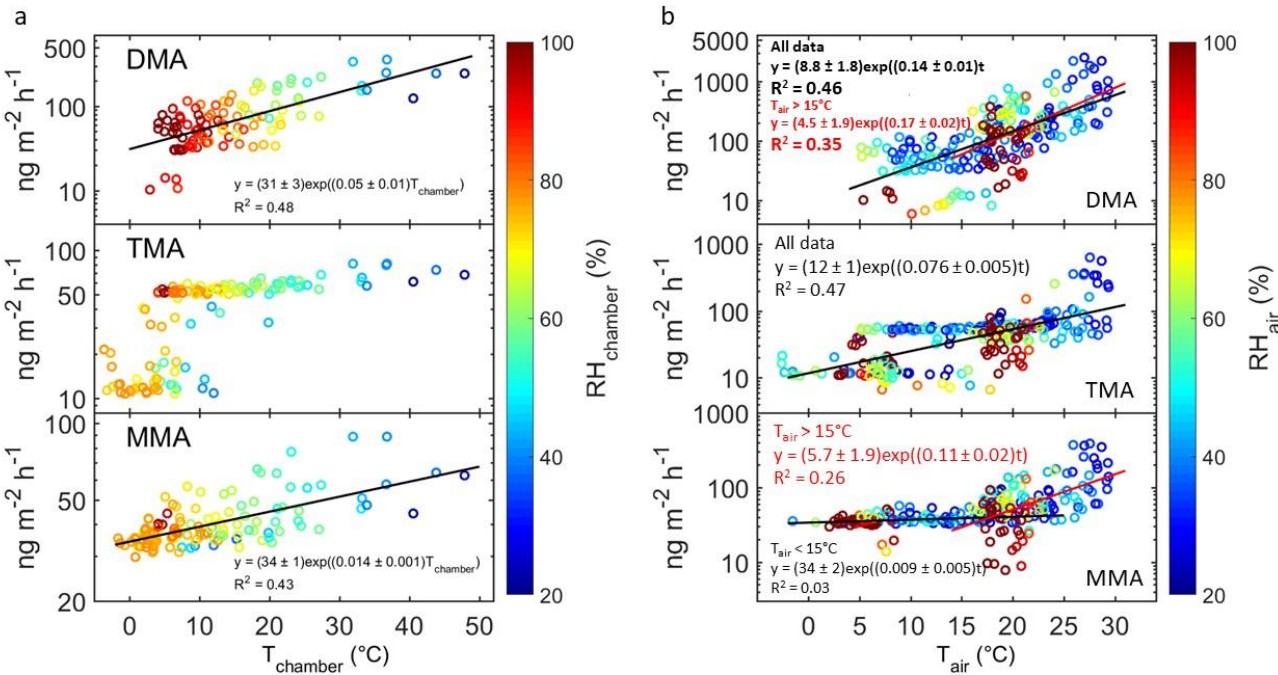

**Figure 11: Emissions of three amines as a function of a) chamber temperature ($T_{chamber}$) and relative humidity ($RH_{chamber}$) and b) outdoor air temperature ($T_{air}$) and relative humidity ($RH_{air}$). Note that in a) there are data only until 28[th] May, due to the malfunctioning of the RH sensor in the chamber, and the highest emissions were observed in July. Note also the different scales in both x and y axes in a) and b).**





### 3.3 Effect of environmental parameters on emission rates

In Figure 9, we can see that even though the maximum chamber temperatures were similar in both May and July, the emission rates and daily emission potentials were much higher in July. In May, the ambient air and soil surface were colder and the soil surface more humid than in July (Table 2). The colder and moister soil surface could explain the lower emissions in May. We compared emission rates of DMA, TMA and MMA to soil surface temperature ($T_{soil}$) and soil surface humidity (SH) (Fig. 12) and found that for higher $T_{soil}$ and lower SH, emission rates were higher. Such dependence was not found for the other amines

and guanidine. The PAR values were also generally lower in May than in July.

The lowest amine emission rates were measured in April and September. Even though chamber, air and soil temperatures were lower and the soil moisture higher in April, the emissions were higher than in September. In April, our measurement chamber was located on melting snow, and this could have been an additional source of amines (Bigg, 2001). Generally, the emission rates were lower during nights, when the chamber, air and soil surface temperature were also lower.


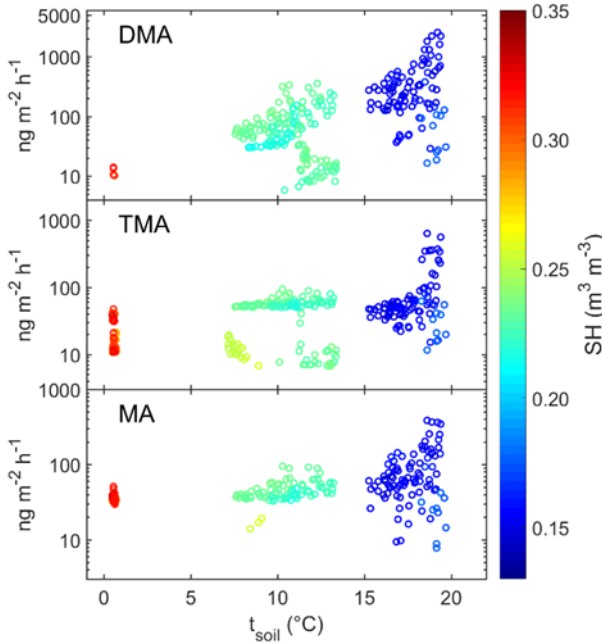

**Figure 12: Emission rates of three amines in the chamber as a function of soil surface temperature ($t_{soil}$, °C) and soil surface humidity (SH, m³ m⁻³) (colour-coded).**






For guanidine, the highest emissions were measured in April during the snow-melting period and lowest in September. No clear explanation for the guanidine emission rate behaviour could be found from the chamber or environmental conditions.

**Table 2. Average chamber and ambient conditions during the measurement periods.**

|  | April | May | July | September |
|---|---|---|---|---|
| $T_{Chamber}$ (°C) | 4.0 | 18.4 | 24.3 | 12.5 |
| $RH_{Chamber}$ (%) | 83 | 79.10 | NaN | NaN |
| PAR ($\mu mol\ m^{-2}\ s^{-1}$) | 160 | 240 | 292 | 102 |
| $T_{Air}$ (°C) | 5.8 | 15.1 | 22.1 | 11.5 |
| $T_{Soil}$ (°C) | 0.6 | 10.2 | 17.7 | 10.9 |
| Surface soil water content ($m^3\ m^{-3}$) | 0.30 | 0.23 | 0.16 | 0.24 |

### 3.4 Error sources in the measurements

Amines are difficult to detect under field conditions, because they are emitted in trace amounts and are highly reactive, so they can be part of chemical reactions and hence removed before they have been sampled and analysed. The recoveries of the

chamber system for MMA, DMA and TMA were only 19%, 29% and 24%, respectively; for guanidine we did not test it. The sampling line was also long (12 m), so losses to the walls were likely, even though we tried to minimize them by heating the inlet tubing. All this suggests, that the observed emission rates are potentially underestimating the true emission rates. The sampling time of MARGA-MS is 1 h, so the results represent cumulative emissions over that time, but we chose the instrument because we could use it to separate various amines with the same mass. The potential challenges and limitations of MARGA-

MS were further discussed in our previous article (Hemmilä et al., 2018).

During the measurements, the temperatures inside the chamber commonly increase, especially if the sampling time is long and the chamber is in direct sunlight. For isoprenoids, increasing the temperature affects the volatility of the compounds and, hence, causes overestimations of their flux rates (Niinemets et al., 2011), which could also be the case for amines. Even though the $T_{chamber}$ was usually very close to the ambient temperature (median difference being 1.4 °C and 66% of the time <

2 °C), in May and July it was very high, especially in the afternoon, maximum difference being 29 °C and 21 °C in May and July, respectively. Obviously, the high temperature inside chamber may enhance amine emissions. However, the observed temperature dependency was in general clearly milder than in the case of isoprenoids (Guenther et al., 2012), so the effect of temperature increase is less pronounced in amines than in isoprenoids. Our emission potential estimates (Table 1) are also independent of temperature. The temperature range in our material is very wide (e.g. Fig. 11) which increases the reliability

of emission potential estimation. In summary, even though our method likely overestimates the emission rates during summer afternoons, in general the effect of the losses is clearly stronger source of underestimation it is a source of overestimation.



However, high-latitude microclimates have been observed temperatures as high as 15 °C above the ambient air temperature and reach values exceeding 30 °C (Rinnan et al., 2014), but the phenomenon in boreal forests the phenomena is probably not as strong.

## 4 Conclusions

In situ amine and guanidine boreal forest floor emissions were measured at the SMEAR II station (Hyytiälä, Finland) in April, May, July and September 2018, about one week per month with time resolution of 1 h. The recovery of emission measurements was tested semi-quantitatively for MMA, DMA and TMA, and the results were 19%, 29% and 24%, respectively. Based on this experiment, the true emission rates are probably four times higher than those presented here.

MMA, DMA and TMA showed maximum emission rates in July, but the highest emission rates for guanidine were already measured in April when the snow was melting. The MMA, DMA and TMA emission rates showed wide diurnal variation, especially in July, with maxima in the afternoon. The DEA emission rates were generally low, showing clear diurnal cycle only in July.

The temperature dependence of emissions was examined; we found a clear correlation between the emission rates and chamber temperature. Based on the assumption of delay between the chamber headspace temperature and soil temperature, and observed delay between emission rate and chamber headspace temperature, the soil temperature is likely the primary environmental control for the DMA and guanidine emissions. TMA showed similar pattern, but emission rates correlated with chamber temperature measured at the same time. The emissions from snow or vegetation rather than from soil could explain this. The emission rates of MMA, DMA, TMA and DEA were highest, when soil surface temperature was high and soil surface humidity low.

The laboratory work has shown that amines are a crucial component in aerosol formation, particularly through their capacity to stabilize acidic clusters during new particle formation (Kurtén et al., 2008; Petäjä et al., 2011; Almeida et al, 2013; Kulmala et al., 2013). Therefore, the results on the amine emissions from the boreal forest soil obtained in this study can be utilized e.g. in scaling up the climatic role of aerosol formation as the amines are important for the nanoparticle formation and their presence in the boreal environment can enhance aerosol formation and growth rates in this environment.

*Data availability*. The data sets can be accessed by contacting the corresponding author.

*Acknowledgements.* The research was supported by the Academy of Finland via the Academy Research Fellow project (Academy of Finland, project 275608) and via Center of Excellence in Atmospheric Sciences (project no. 272041) and European Research Council via ATM-GTP 266 (742206). This research has also received funding from Academy of Finland (project no. 316114 & 315203) as well as the Doctoral Programme in Atmospheric Sciences at the University of Helsinki.



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
