# Peer review of "Amine and guanidine emissions from a boreal forest floor"

_Atmospheric Chemistry and Physics, 2019_

## Referee Comment (RC1) · Anonymous Referee #1 · 8 Mar 2020

**General Comments**

This manuscript has two fatal flaws which make it unacceptable for publication in Atmospheric Chemistry and Physics. Significant additional methodological and sampling detail must be added to the manuscript and a thorough review of the literature on the bi-directional exchange of ammonia are crucial to making this work acceptable for publication. As the manuscript currently stands, there is insufficient information to determine whether the diurnal emission patterns, or longer monthly rates, in the observed amines are simply a result of thermodynamic adsorption partitioning off the chamber, tubing, and inlet surfaces of the MARGA-MS instrument versus true environmental phenomena. The experimental data from the chamber transmission efficiency suggest the reported hourly observations could easily be driven by temperature-dependent ad-

sorption effects and it is, in fact, a common issue reported from decades of ammonia (NH3) measurements. It would not be surprising for these amine (NR3) measurements to be affected in the same way, but the lack of detail regarding the setup and validation of the chambers make it impossible to assess. Second, the discussion regarding the underlying mechanisms of NR3 emissions from soil pore water and/or surfaces suffers similarly from a decided lack of literature review regarding well-established effective Henry's law partitioning and compensation points for NH3. The chemical nature of NR3 are nearly synonymous with NH3 and this work ignores the context of our existing understanding entirely. This has resulted in discussion of NR3 volatility under environmental conditions as if they are evaporating from pure liquid amines, which is certainly not the case. At a minimum, resolving these two issues requires a major revisiting of the collected data sets and their interpretation. In addition, there are many major issues that also need to be addressed.

Major Comments

1. The operation of the MARGA-MS, its data validation and quality control practices are lacking. The Authors regularly reference their prior work. If such heavy reliance on this prior work is critical to the reported measurements, surely some extended discussion of the details applied in this field work are warranted? For example, calibrations of the system are poorly defined. Were these gas phase additions to the inlet? Aqueous standards? What ambient concentration range do the calibration standards correspond to? How were detection limits determined for this set of field measurements? Use of wet chemical techniques under field conditions has been regularly reported to require inlet overflow with zero air in order to appropriately correct background contamination in water sources, reagents, and other components due to the challenge in achieving lab-level cleanliness in the field. Much of the data presented in the Figures of the manuscript look to have some sort of background amount present and it is not clear whether any corrections for systematic contamination of the system have been made. The Authors must address in detail, in this manuscript, for this set of field measurements, how they performed field blanks, calibrations, and any other corrections to their data. This would then allow a sound assessment of how the detection limits were determined. It is currently unclear if the Authors are relying on a prior determination of detection limits or whether they were determined for this sampling period through the analysis of negative controls (e.g. sampled zero air through inlet lines or a control chamber). The necessity of determining field blanks for each field measurement period has been clearly demonstrated in prior work in real time wet chemical sampling of NH3 and NR3 for separation by ion chromatography (Von Bobrutzki et al., 2010; Markovic et al., 2012; VandenBoer et al., 2011, 2012). Some MARGA systems use an internal standard of LiBr in their denuder solutions to correct for systematic error in the volume of samples and instrumental variability in measurements. Given the presentation of 'zig zag' data here, it does not seem like internal standards were used to correct for systematic issues in the quantification of NR3. It is extremely surprising to see the Authors present field data with such a glaring systematic issue that is easily corrected based on the operating principles of the MARGA. In fact, they state that this is a known feature of MARGA data that they collect, but how is this not possible to correct? Such deliberate use of low-quality data fails to establish confidence that the reported mixing ratios are reliable.

2. The siting and operation approach of the dynamic flux chamber(s) are not provided at all. How many chambers were deployed? What was the flow of air through the chambers? Was this a larger flow than what was sampled by the MARGA? What is the residence time of a molecule of NR3 emitted into the chamber prior to being removed by sampling? Rough dimensions of the chamber are stated, but a volume is not given. These are critical to understand the potential quality of the data presented from the use of these chamber(s). No data are provided either for the efficacy of the oxalic acid filter to remove amines flowing from ambient air into the chamber. Typically, citric acid or phosphorous acid is used for such collections (Key et al., 2011), so it is valuable to demonstrate that a chamber fitted with the oxalic acid filter, placed on an inert surface, does not allow any amines into the chamber. Further to this, there is no rationale

for the siting of the chamber given. If multiple chambers were used, was this to get representative emission values? If only one chamber was used, was it located on a surface that was thought likely to emit amines? If not, how was the location selected? Please be clear if the location of the chamber was simply to determine if it was possible to detect amine surface emissions without any intention to achieve representativeness of amine emission rates. Be clear in updating the discussion of literature comparisons as well if this is the case. Finally, were the chambers kept closed at all times or closed only for sample collection? If they were deployed in an array, were they operated on a schedule suitable for reactive gases at parts per trillion levels (pptv) (Pape et al., 2009; Plake et al., 2015)? Do the Authors have any proof that the chambers operated soundly for the measurement of $CO_2$ or $H_2O$ surface emissions? This would lend credibility to the operating parameters of the chamber.

The Authors state that they tested several materials for the transfer of $NR_3$ from a permeation oven to the MARGA, but do not provide any of the data to demonstrate the response time of the setup (i.e. 10-90 %) to increases in $NR_3$ mixing ratios, nor the temporal nature of desorption effects (i.e. 90-10 %) when switching from measuring $NR_3$ to zero air, to justify their selection. The lack of this data makes it impossible to determine whether the subsequent field observations are valid. The Authors have not previously reported that they have established such a technique, so it is critical that they thoroughly validate their approach in this work. Once through the chamber(s), the amines then transit 12 m of FEP tubing. Again, flow/residence time are not given, neither is the temperature at which the lines were heated, so it is impossible to determine the extent to which inlet effects could impact the reported observations or result in lag times that could explain the offset time between $NR_3$ emissions and T-chamber (Von Bobrutzki et al., 2010; Deming et al., 2019; Ellis et al., 2010; Liu et al., 2019; Moravek et al., 2019; Pollack et al., 2019). The reported surface losses of 70-80 % for the tested $NR_3$ (MMA, DMA, and TMA) is clearly indicative that subsequent release back into the sample flow is possible, a process which is entirely reasonable to expect will demonstrate T-dependent reversibility.

Detection limits (at the 3-sigma level) for the MARGA-MS chambers are reported, but how they were determined are not. Given the low NR3 transmission efficiency, the chamber methodology must have at least the same factor of 4-5 (or higher) increased detection limits over those of the MARGA-MS system alone. These detection limits need to be depicted as horizontal lines on all plots where emission rates are reported.

3. Chamber partitioning and MARGA denuder-to-injection duration could easily explain the time lag of 2 hours needed to explain the relationship to T-chamber. Given that the Authors have not corrected their MARGA-MS measurements for known systematic measurement biases, despite stating that they have used an internal standard to track such bias, and opted instead to publish 'zig zag' data, consideration of lag time in emitted amines reaching the detector seem unlikely to have been characterized. The origin of this time lag could have easily been established or rejected if the transmission fraction results from the chambers had been provided in the Supporting Information document for this work. There is sufficient literature precedent for such a time-lag for reduced nitrogen species (NH3 and NR3) in biasing temporal trends in wet chemical sampling instrumentation (Von Bobrutzki et al., 2010; Deming et al., 2019; Ellis et al., 2011; Markovic et al., 2012).

Detailed Comments Lines 33-34: Should be 'decarboxylation'

Line 35: This is the first instance of incorrect understanding of the exchange of NR3 from surfaces under environmental conditions. Water is present on all surfaces and NR3 in an aqueous environment will rapidly establish an equilibrium with their protonated form (NR3H+) according to their weak base character. Therefore, their exchange between environmental surfaces, soil pore water, and plant apoplastic fluid is determined by the product of the NR3 volatility (Henry's Law constant) and its base dissociation constant (Kb).

Lines 49-50: What is the volatility of guanidine? Has its volatility been previously measured? What about its Kb values? Such data could provide easy insight into whether it

will or will not be possible to exchange between environmental surfaces and the over-lying atmosphere. The Authors then indicate that guanidine has been detected previously in Alaska, but fail to report the matrix it was isolated from. Is this gaseous, in soil water, something else? It is hard to follow the relevance of this statement otherwise.

Lines 117-118: MMA incorrectly interpreted as a pure liquid under environmental conditions. Revise interpretation using effecting partitioning concept and compensation point. Lines 120-122: Except, if your observations following peak emission are biased high as the NR3 repartition into cleaner air, then the integrated emission rate is correct. In the absence of the chamber transmission data to demonstrate the degree of reversibility of NR3 partitioning to surfaces of the sampling setup, it is impossible to accurately determine the degree of bias in emission rates or potential time offsets.

Lines 124-125 and Figure 3: This justification for presenting systematically biased data for measurements of pptv levels of amines is absurd. The Authors having been allowed to use this erroneous approach in prior work does not validate its continued use. It must be corrected for here.

Figures 1 and 2: There is no data presented for the months of June and August. An explanation for the gap in the observations is necessary to present in the methods section of the manuscript and to re-iterate in the figure caption.

The continuity of the datasets collected is also not reported. In the prior work with the MARGA-MS by these same Authors, there were major limitations in data continuity from this platform due to instrument failure. The number of data points collected in each month are critical to report. Without the number of measurements being used to compare between months, it is not possible to discern the potential for bias due to sampling limitations.

Figure 3: Alphabetic labels should be placed outside of each panel. The legend needs to be presented only once (or better yet, only described in the caption text). It would be better to also show the PAR measurements on these plots. In the caption, the

Authors state that the error bars show the standard deviation: is this one, two, or three standard deviations? Here and moving forward in the manuscript, the exact quantity for error needs to be described.

In the caption the Authors state 'in April the soil surface humidity did not change during the measurement period, so no data are shown'. The reasoning here is very confusing. Simply plot the data so Readers can compare across all observations easily. One should not have to go to the figure caption to determine that the missing points would simply be present in a straight line if plotted.

The last line in the figure caption refers to what must be a prior version of the figure which had emission rates presented. They are no longer here. Delete this reference to emission rates.

Figure 4: All of the y-axes should have values starting at zero with the detection limits for each amine emission rate plotted as a horizontal dashed line on each panel. This will allow easy interpretation of when emissions were above the method detection limits. Panel d suggests that there is an uncorrected background of MMA in the field system. In the absence of an adequate description of the regularity of measuring instrument backgrounds and correction of the collected data, it is not possible to ascertain whether these are real measurements or an instrument background.

Lines 167-171: The limited depth of understanding on soil exchange and ambient mixing ratios of NR3 is clearly demonstrated in this discussion. Units for the DEA compensation point are not given, but are presumably pptv? Further to this, soil pH for the region must have been previously characterized? Surely an estimate of the expected direction of exchange based on the concept of compensation point can be made. Here the Authors are comparing their measurements against prior reports where effective partitioning coefficients of NR3 were calculated, yet here the interpretation is limited to volatility or boiling points instead. This must be corrected. This prior work also suggests that amines were emitted from soils which should not emit them, based on the

[Figure]

concept of compensation point. This is a very important detail that goes unacknowledged in the discussion. Boreal soils, particularly for podzols, are very acidic. Release of basic species, like NR3, would be highly unfavorable if they are originating from the soil pore water with a pH set by the bulk pH measured by standard techniques. This suggests that NR3 must originate from a different process or that the concepts set by bulk soil properties are insufficient to describe how NR3 are emitted from Boreal soils. In either case, the observations warrant a better discussion on this point since compensation point theory has been largely successful for NH3.

Figure 5: Again, all y-axes should start from a value of zero. In the caption, the description of error bars is incorrect. They show the variance of the measurements about the mean value, not the deviation between measured values. The September measurements in panel d, again, suggest there is an uncorrected background of DMA, as there also seems to be for MMA.

Figure 6: Same incorrect error bar description as in Figure 5.

Figure 7: In panel b the variance in the data is much higher than in the other panels. Why is this? Surely this warrants some explanation in the caption?

Lines 191-192: This interpretation shows very little though has been put into the discussion of these observations, or the chemical properties governing the exchange of volatile bases from environmental surfaces in general. For guanidine emissions from snowpack, under what conditions could these observations justifiably occur? Why were snow samples not collected and analyzed when these observations were detected? The same could be said for the remainder of the reported amines and their detection in soil pore water during the different months when observations were made.

Figure 8: The difference in the variability of the measurements reported between each month suggest that the instrument and/or method detection limits are not constant and may be changing over time. The Authors need to comment on this in addition to depicting the detection limits for each observation period with a horizontal line on each

emission rate plot.

Lines 215-221: Why are monoterpene emissions relevant to soil processes? Are monoterpene emissions a result of physical, chemical, or biological processes? Do they arise from plants, microbes, fungi? Why are monoterpene emissions appropriate to discuss in order to interpret the amine emission observations? The purpose for presenting this section of discussion is unclear and needs to be revised for relevance or removed.

Lines 226-227: Here the Authors acknowledge that amines are water-soluble and can be removed from surfaces and soil pore water in rain. This is at odds with all prior discussion of amine volatilization and boiling points. Consistency in interpretation of NR3 fate and partitioning needs to be present throughout this work.

Figure 9: Are the points plotted at zero? Or are these below the detection limit? Are these periods when blanks were measured? It is not possible to tell given the sampling description. Have these time series been adjusted for the 2-hour offset between chamber temperature? The caption does not clarify. The lag in the rise and fall of each NR3 species suggests that there are significant inlet effects downstream of the chamber.

Figure 10: The text states that guanidine is supposed be on this plot, yet it seems to be absent? Please add it and remove the DMA 'same time' data. The caption also suggests that only a single day of data was analyzed to investigate this T-dependent relationship. This seems unnecessarily restricted given that there are at least four months of data to parse for a robust explanation of the observed relationship. These findings highlight the weakness of the sampling approach and its lack of validation against sampling artifacts, specifically against T-driven partitioning effects. If anything, this plot emphasizes that it is quite possible that the reported diurnal NR3 emission trends are only repartitioned species from the chamber and sampling line surfaces.

Table 1: What are the columns labeled 'R2 without temperature move' supposed to represent? Those without the 2-hour time adjustment? Why bother with reporting this?

Simply state in the caption that the R2 is presented for 2-hour adjusted temperature at 30 C in the caption and get rid of these columns.

Lines 247-250: The order of these k values should be compared to the effective partitioning coefficients for these NR3 to see if they are consistent with their expected behavior under environmental conditions.

Figure 11: The left panels are strongly suggestive of reversible chamber wall partitioning instead of direct soil emission, especially with how they are sorting clearly by the relative humidity of the chamber. Without adequate quantitation of surface effects in the experimental system, the temporal nature and magnitude of the NR3 emissions cannot be accurately determined.

References

Von Bobrutzki, K., Braban, C. F., Famulari, D., Jones, S. K., Blackall, T., Smith, T. E. L., Blom, M., Coe, H., Gallagher, M., Ghalaieny, M., McGillen, M. R., Percival, C. J., Whitehead, J. D., Ellis, R., Murphy, J., Mohacsi, A., Pogany, A., Junninen, H., Rantanen, S., Sutton, M. A. and Nemitz, E.: Field inter-comparison of eleven atmospheric ammonia measurement techniques, Atmos. Meas. Tech., 3(1), 91–112, doi:10.5194/amt-3-91-2010, 2010.

Deming, B. L., Pagonis, D., Liu, X., Day, D. A., Talukdar, R., Krechmer, J. E., De Gouw, J. A., Jimenez, J. L. and Ziemann, P. J.: Measurements of delays of gas-phase compounds in a wide variety of tubing materials due to gas-wall interactions, Atmos. Meas. Tech., 12(6), 3453–3461, doi:10.5194/amt-12-3453-2019, 2019.

Ellis, R. A., Murphy, J. G., Pattey, E., Van Haarlem, R., O'Brien, J. M. and Herndon, S. C.: Characterizing a Quantum Cascade Tunable Infrared Laser Differential Absorption Spectrometer (QC-TILDAS) for measurements of atmospheric ammonia, Atmos. Meas. Tech., 3(2), 397–406, doi:10.5194/amt-3-397-2010, 2010.

Ellis, R. A., Murphy, J. G., Markovic, M. Z., Vandenboer, T. C., Makar, P. A., Brook, J.

and Mihele, C.: The influence of gas-particle partitioning and surface-atmosphere exchange on ammonia during BAQS-Met, Atmos. Chem. Phys., 11(1), doi:10.5194/acp-11-133-2011, 2011.

Key, D., Stihle, J., Petit, J. E., Bonnet, C., Depernon, L., Liu, O., Kennedy, S., Latimer, R., Burgoyne, M., Wanger, D., Webster, A., Casunuran, S., Hidalgo, S., Thomas, M., Moss, J. A. and Baum, M. M.: Integrated method for the measurement of trace nitrogenous atmospheric bases, Atmos. Meas. Tech., 4(12), 2795–2807, doi:10.5194/amt-4-2795-2011, 2011.

Liu, X., Deming, B., Pagonis, D., Day, D. A., Palm, B. B., Talukdar, R., Roberts, J. M., Veres, P. R., Krechmer, J. E., Thornton, J. A., De Gouw, J. A., Ziemann, P. J. and Jimenez, J. L.: Effects of gas-wall interactions on measurements of semivolatile compounds and small polar molecules, Atmos. Meas. Tech., 12(6), 3137–3149, doi:10.5194/amt-12-3137-2019, 2019.

Markovic, M. Z., Vandenboer, T. C. and Murphy, J. G.: Characterization and optimization of an online system for the simultaneous measurement of atmospheric water-soluble constituents in the gas and particle phases, J. Environ. Monit., 14(7), doi:10.1039/c2em00004k, 2012.

Moravek, A., Singh, S., Pattey, E., Pelletier, L. and Murphy, J. G.: Measurements and quality control of ammonia eddy covariance fluxes: A new strategy for high frequency attenuation correction, Atmos. Meas. Tech. Discuss., (June), 1–30, doi:10.5194/amt-2019-193, 2019.

Pape, L., Ammann, C., Nyfeler-Brunner, A., Spirig, C., Hens, K. and Meixner, F. X.: An automated dynamic chamber system for surface exchange measurement of non-reactive and reactive trace gases of grassland ecosystems, Biogeosciences, 6(3), 405–429, doi:10.5194/bg-6-405-2009, 2009.

Plake, D., Stella, P., Moravek, A., Mayer, J. C., Ammann, C., Held, A. and

Trebs, I.: Comparison of ozone deposition measured with the dynamic chamber and the eddy covariance method, Agric. For. Meteorol., 206, 97–112, doi:10.1016/j.agrformet.2015.02.014, 2015.

Pollack, I. B., Lindaas, J., Robert Roscioli, J., Agnese, M., Permar, W., Hu, L. and Fischer, E. V.: Evaluation of ambient ammonia measurements from a research aircraft using a closed-path QC-TILDAS operated with active continuous passivation, Atmos. Meas. Tech., 12(7), 3717–3742, doi:10.5194/amt-12-3717-2019, 2019.

VandenBoer, T. C., Petroff, A., Markovic, M. Z. and Murphy, J. G.: Size distribution of alkyl amines in continental particulate matter and their online detection in the gas and particle phase, Atmos. Chem. Phys., 11(9), doi:10.5194/acp-11-4319-2011, 2011.

VandenBoer, T. C., Markovic, M. Z., Petroff, A., Czar, M. F., Borduas, N. and Murphy, J. G.: Ion chromatographic separation and quantitation of alkyl methylamines and ethylamines in atmospheric gas and particulate matter using preconcentration and suppressed conductivity detection, J. Chromatogr. A, 1252, doi:10.1016/j.chroma.2012.06.062, 2012.

---

## Referee Comment (RC2) · Anonymous Referee #2 · 3 Apr 2020

This manuscript describes a set of measurements made using a MARGA-MS instrument at a boreal forest site during four different months of the year. The MARGA-MS was sampling the outflow of a closed dynamic chamber, which was intended to capture emissions of amines and guanidine from the forest soil. There are relatively few reports of amine emission rates from soil so new information would be valuable. Unfortunately, there are major problems with the experimental design that mean that the data cannot be trusted quantitatively. Some of these issues (described below) even call into question the trends and relative relationships that are reported, meaning that the interpretations are very unreliable. As such, I do not think this paper should be published in ACP.

The authors note very substantial adsorptive and/or reactive losses of the amines dur-

ing quality control tests, even with the best choice of material. The authors make the decision to not account for these losses in calculating the emission fluxes. While this is a questionable decision, it might be overcome if we thought that the losses were consistent over the sampling campaign. The issue is that the losses are likely dependent on ambient conditions (temperature, humidity, insolation) and probably also on the magnitude of emissions. Therefore, it becomes difficult to trust their results, even in a relative or semi-quantitative way. A substantial fraction of the manuscript is devoted to examining how the emissions depend on environmental factors, but it may be that the adsorptive losses are what depends on these factors. From my understanding of the data, there is not way that these issues can be disentangled.

There is another issue in the design of the sampling strategy that undermines the value of the data. The use of amine-free air as the inflow to the chamber means that the emissions of these compounds may not be relevant for ambient conditions. The bidirectional framework that is typically used to explain atmosphere-ecosystem fluxes of ammonia should also apply to bases like amines and guanidine. In this framework, the fluxes are understood to result from the difference between the ambient concentration and the compensation point established by the aqueous (or soil, vegetation) reservoirs of the compound. By setting the initial ambient concentration to zero, the net fluxes are going to be higher (and possibly of a different sign) than they would be in the natural environment. Further, when used for flux measurements, these closed chambers are typically left open for a substantial fraction of each day to ensure that the ground surface enclosed by the chamber has the ability to interact with the overlying atmosphere. Based on the way in which the measurement are presented, it seems that the chamber was closed for many days on end. If this is true, this also undermines the applicability of this system to the natural and undisturbed ecosystem.